# Profiling immunoglobulin repertoires across multiple human tissues using RNA sequencing

Igor Mandric[1], Jeremy Rotman[1,2], Harry Taegyun Yang [1,3], Nicolas Strauli[4], Dennis J. Montoya[5], William Van Der Wey[1], Jiem R. Ronas[6], Benjamin Statz[1], Douglas Yao[5,7], Velislava Petrova[8], Alex Zelikovsky [9,10], Roberto Spreafico[11], Sagiv Shifman [12,19], Noah Zaitlen [13,19], Maura Rossetti[14,19], K. Mark Ansel [15,19], Eleazar Eskin[1,16,17,19] & Serghei Mangul [1,2,11,18,19 ✉]

Profiling immunoglobulin (Ig) receptor repertoires with specialized assays can be cost-ineffective and time-consuming. Here we report ImReP, a computational method for rapid and accurate profiling of the Ig repertoire, including the complementary-determining region 3 (CDR3), using regular RNA sequencing data such as those from 8,555 samples across 53 tissues types from 544 individuals in the Genotype-Tissue Expression (GTEx v6) project. Using ImReP and GTEx v6 data, we generate a collection of 3.6 million Ig sequences, termed the atlas of immunoglobulin repertoires (TAIR), across a broad range of tissue types that often do not have reported Ig repertoires information. Moreover, the flow of Ig clonotypes and inter-tissue repertoire similarities across immune-related tissues are also evaluated. In summary, TAIR is one of the largest collections of CDR3 sequences and tissue types, and should serve as an important resource for studying immunological diseases.

[1] Department of Computer Science, University of California, Los Angeles, 404 Westwood Plaza, Los Angeles, CA 90095, USA. [2] Department of Clinical Pharmacy, School of Pharmacy, University of Southern California, 1540 Alcazar Street, Los Angeles, CA 90033, USA. [3] Bioinformatics Interdepartmental Ph. D. Program, University of California, Los Angeles, 611 Charles E. Young Drive East, Los Angeles, CA 90095-1570, USA. [4] Biomedical Sciences Graduate Program, University of California, San Francisco, 1675 Owens Street, Suite 310, San Francisco, CA 94143-0523, USA. [5] Department of Molecular, Cell, and Developmental Biology, University of California, Los Angeles, 610 Charles E. Young Drive South, Los Angeles, CA 90095, USA. [6] Department of Microbiology, Immunology, and Molecular Genetics, University of California, Los Angeles, 609 Charles E. Young Drive East, Los Angeles, CA 90095, USA. [7] Program in Bioinformatics and Integrative Genomics, Harvard Medical School, 10 Shattuck Street, Suite 514, Boston, MA 02115, USA. [8] Wellcome Trust Sanger Institute, Wellcome Genome Campus, Hinxton, Cambridge CB10 1SA, UK. [9] Department of Computer Science, Georgia State University, 33 Gilmer Street SE, Atlanta, GA 30303, USA. [10] The Laboratory of Bioinformatics, I.M. Sechenov First Moscow State Medical University, Moscow 119991, Russia. [11] Institute for Quantitative and Computational Biosciences, University of California, Los Angeles, 611 Charles E. Young Drive East, Los Angeles, CA 90095, USA. [12] Department of Genetics, The Institute of Life Sciences, The Hebrew University of Jerusalem, Jerusalem 9190401, Israel. [13] Department of Medicine, University of California, San Francisco, 533 Parnassus Avenue, San Francisco, CA 94143, USA. [14] Immunogenetics Center, Department of Pathology and Laboratory Medicine, David Geffen School of Medicine, University of California, Los Angeles, 1000 Veteran Avenue, Los Angeles, CA 90095-1652, USA. [15] Sandler Asthma Basic Research Center, Department of Microbiology and Immunology, University of California, San Francisco, 513 Parnassus Avenue, San Francisco, CA 94143-0414, USA. [16] Department of Human Genetics, David Geffen School of Medicine at UCLA, 695 Charles E. Young Drive South, Box 708822, Los Angeles, CA 90095, USA. [17] Department of Computational Medicine, David Geffen School of Medicine at UCLA, 73-235 CHS, Los Angeles, CA 90095, USA. [18] Quantitative and Computational Biology, University of Southern California, Los Angeles, CA 90089, USA. [19]These authors jointly supervised this work: Sagiv Shifman, Noah Zaitlen, Maura Rossetti, K. Mark Ansel, Eleazar Eskin, Serghei Mangul. ✉email: serghei.mangul@gmail.com

A key function of the adaptive immune system is to mount protective memory responses to a given antigen. B cells recognize their specific antigens through immunoglobulins (Ig), surface antigen receptors, which are unique to each cell and its progeny. A typical Ig repertoire is composed of one immunoglobulin heavy chain (IGH) and two light chains, κ (IGK) and λ (IGL). Igs are diversified through somatic recombination, a process that randomly combines variable (V), diversity (D), and joining (J) gene segments, and inserts or deletes non-templated bases at the recombination junctions[1] (Fig. 1a). The resulting DNA sequences are then translated into antigen receptor proteins. This process enables the Ig repertoire to develop astonishing diversity of antigen receptors from any given individual, with >$10^{13}$ theoretically possible distinct Ig receptors[1]. Ig repertoire diversity is key for an individual's immune system to confer protection against a wide variety of potential pathogens[2]. In addition, upon activation of a B cell, somatic hypermutation further diversifies Ig in their variable region. These changes are mostly single-base substitutions occurring at extremely high rates —somatic hypermutation can undergo $10^{-5}$ to $10^{-3}$ mutations per base pair per generation[3]. Isotype switching is another mechanism that contributes to B-cell functional diversity. Here, antigen specificity remains unchanged, while the heavy chain VDJ regions join with different constant (C) regions, such as IgG, IgA, or IgE isotypes, and alter the immunological properties of Igs. The pairing of heavy and light chains that occurs in polyclonally activated B cells chains is another mechanism that increases Ig diversity.

High-throughput technologies enable remarkable levels of accuracy when profiling the Ig repertoires. Commonly used assay-based approaches to RNA sequencing (RNA-Seq) provide a detailed view of the adaptive immune system by leveraging the deep sequencing of amplified DNA or RNA from the variable region of the Ig locus (BCR-Seq)[4–6]. Those technologies are usually restricted to one chain, with the majority of studies focusing on the heavy chain of the Ig repertoire. Recent studies[2] have successfully applied assay-based approaches to characterize the immune repertoire of peripheral blood. However, little is known about the immunological repertoires of other human tissues, including barrier tissues such as skin and mucosae. Studies involving assay-based protocols usually have small sample sizes, thus limiting analysis of intra-individual variation of immunological receptors across diverse human tissues.

RNA-Seq traditionally uses the reads mapped onto human genome references to study the transcriptional landscape of both single cells and entire cellular populations. In contrast to assay-based protocols that produce reads from the amplified variable region of the Ig locus, RNA-Seq is able to capture the entire cellular population of the sample, including B cells. However, due to the repetitive nature of the Ig locus and the extremely high level of diversity in Ig transcripts, most mapping tools are ill-equipped to handle Ig sequences. RNA-Seq was successfully used for analysis of highly clonal leukemic repertoires with high relative quantities of Ig transcripts[5]. Despite this, Ig transcripts often occur in sufficient numbers within the transcriptome of many tissues to characterize their respective Ig repertoires[7]. A number of methods[8–10] were designed to assemble Ig and T-cell receptor repertoires and have been applied across various public RNA-Seq datasets. Existing methods that are capable of assembling Ig repertoires from bulk RNA-Seq data typically produce low-accuracy results (F-score < 0.2).

In this study, we develop ImReP, an alignment-free computational method for rapid and accurate profiling of the Ig repertoire from regular RNA-Seq data. ImReP is capable of efficiently extracting receptor-derived reads from RNA-Seq data and accurately assembling Ig clonotypes, defined as distinct amino acid sequences of complementarity-determining region 3 (CDR3). We demonstrate that bulk RNA-Seq is a suitable technology for measuring the individual adaptive immune repertoire. In our validation experiments, ImRep is able to capture 69% of the immune repertoire obtained by BCR-Seq. Using ImReP, we create a systematic atlas of Ig sequences across a broad range of tissue types, most of which were previously unstudied for Ig repertoires. We also examine the compositional similarities of clonal populations between the tissues to track the flow of Ig clonotypes across immune-related tissues, including secondary lymphoid and organs that encompass mucosal, exocrine, and endocrine sites. Our proposed approach lacks advantages in comparison with performing targeted BCR-Seq; rather, it provides a useful tool for mining large-scale RNA-Seq datasets for the study of Ig receptor repertoires.

## Results

**Existing tools for profiling the Ig receptor repertoire.** A number of tools have previously been developed to reconstruct the Ig receptor repertoire. Repertoire analysis from RNA-Seq data typically starts with mapping the reads to the germline V, D, and J genes that can be obtained from the International ImMuno-GeneTics (IMGT) database[11]. There are three possible read mapping scenarios as follows: (1) the read is entirely mapped to the V gene; (2) the read is entirely mapped to the J gene; (3) the read is partially mapped to the V and J genes simultaneously. Existing methods consider only reads from category (3). These methods use different underlying algorithms to map reads to germline genes. For example, MiXCR[8] relies on an in-house alignment procedure, IgBlast[12] utilizes BLAST with an optimized set of parameters, and IMSEQ[13] uses in-house pairwise alignment between the read sequence and the germline V and J segment sequences.

Following the alignment, MiXCR performs overlapping of previously aligned reads into contigs. The resulting contigs are re-aligned to the V, D, and J genes to verify that the significant portion of non-template N insertions is covered. In contrast to MiXCR, which simultaneously aligns reads to both V, D, and J genes, IgBlast separately aligns the query read to databases composed of V, D, and J genes. IgBlast uses a specific sequence to separately perform alignment; first, the program finds the best V gene hit. Next, IgBlast masks the aligned read region and performs an alignment to the J gene database. In the event of a heavy chain, IgBlast also queries the D gene database for the best hit. Lastly, the software checks that each component in the obtained V(D)J rearrangement originates from the same locus and reports CDR3 sequences and corresponding V(D)J recombinations.

All methods use the definition of CDR3 to determine the boundaries of the CDR3 sequence in each of the reads. The final step in repertoire analysis is to correct the assembled clones for PCR and sequencing errors. To correct these errors, which are introduced during data preparation, MiXCR and IMSEQ cluster the assembled clones and report a consensus sequence for each cluster. IgBlast skips the error correction step and directly outputs inferred CDR3 sequences.

Most methods use alignment or assembly to infer CDR3s and align reads to V and J genes. In contrast, the ImReP procedure provides a match between the read prefix and the read suffix to the prefix of J genes and suffix of V genes, respectively, without a need for alignment. In avoiding alignment, ImReP is able to significantly decrease running time and minimize required computational resources. Average CPU (central processing unit) time reported for ImReP is 44 minutes, a runtime substantially shorter than the average 10 hours required for MiXCR. At peak usage, across all samples, ImReP consumes 3 GB

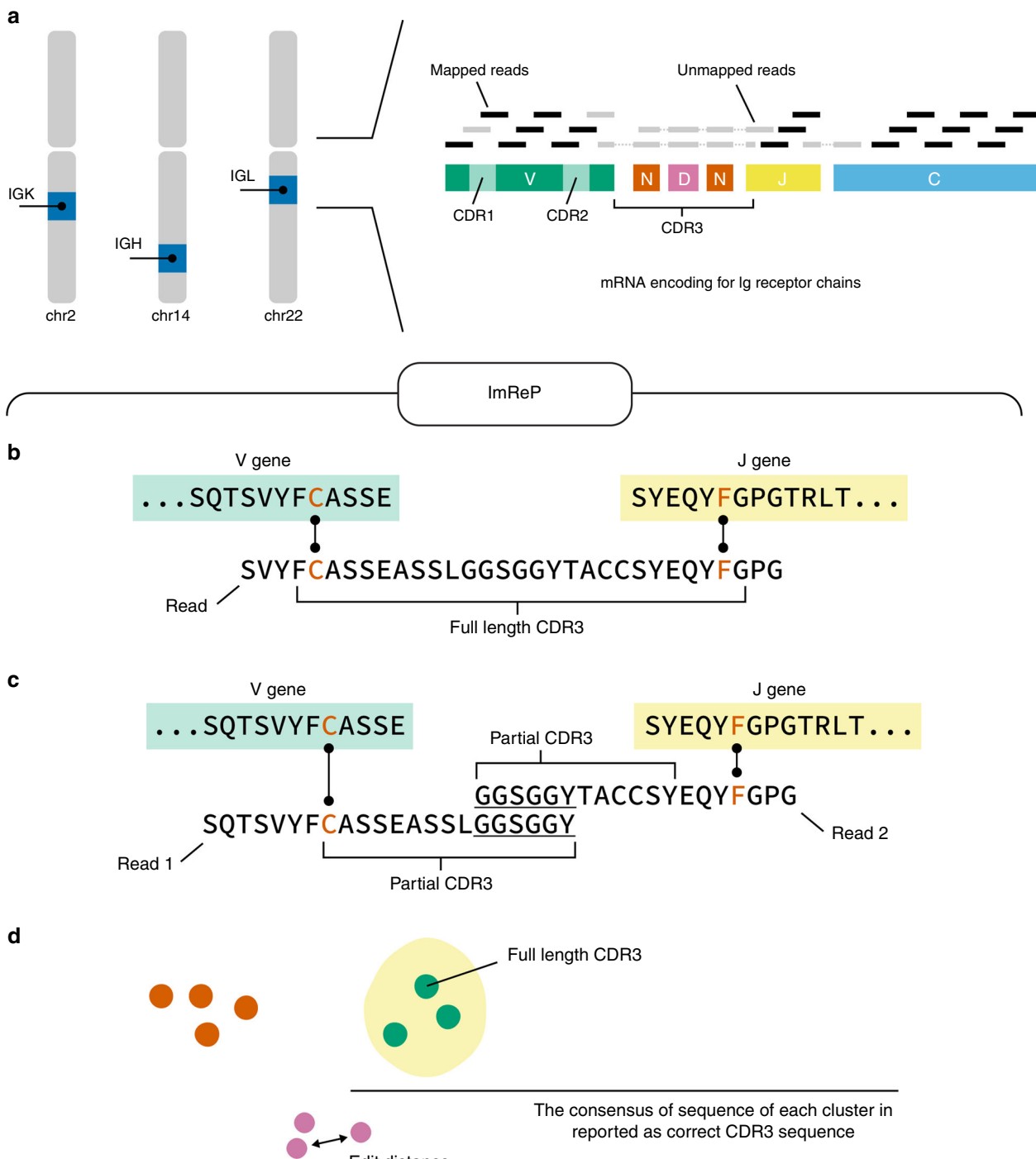

**Fig. 1 Overview of ImReP. a** Schematic representation of human Ig receptor repertoire. Ig repertoire consists of three Ig loci: immunoglobulin heavy locus (IGH), immunoglobulin κ locus (IGK), and immunoglobulin λ locus (IGL). Ig receptors contain multiple variables (V, green color), diversity (D, which is present only in IGH, violet color), joining (J, yellow color), and constant (C, blue color) gene segments. V(D)J gene segments are randomly joined and non-templated bases (N, dark red color) are inserted at the recombination junctions. Reads entirely aligned to Ig genes are inferred from mapped reads (black color). Reads with extensive somatic hypermutations and reads spanning the V(D)J recombination are inferred from the unmapped reads (gray color). Complementarity-determining region 3 (CDR3) is used to identify Ig receptor clonotypes—a group of clones with identical CDR3 amino acid sequences. **b** Alignment-free detection of reads containing full-length CDR3s and simultaneously overlapping V and J genes. Receptor-derived reads spanning V(D)J recombinations are identified from unmapped reads and assembled into the CDR3 sequences. We first scan the amino acid sequences of the read to determine putative CDR3 sequences that are fully contained inside the read. Reads with putative CDR3s are further examined to simultaneously overlap V and J gene segments. The alignment between the read and V and J genes is found by matching the prefix and suffix of the read to match the suffix of V and prefix of J genes, respectively. **c** Match reads containing partial CDR3s and overlapping only V or J genes. In cases where a read contains a partial CDR3 sequence and overlaps with only the V or J gene, we perform the second stage of ImReP. During this stage, we match reads originated from the same CDR3 based on 15 nucleotides overlap. **e** Correcting PCR and sequencing errors via CAST clustering. We further correct PCR and sequencing errors in the assembled CDR3s. ImReP clusters assemble CDR3 into a set of clusters via CAST algorithm. The consensus sequence of each cluster is reported as the correct CDR3 sequence.

of RAM (random access memory), whereas MiXCR requires 10 GB of RAM.

**ImReP is a method for profiling of Ig repertoire**. We apply ImReP to 0.6 trillion RNA-Seq reads (92 Tbp) from 8555 samples to assemble CDR3 sequences of Ig receptors (Supplementary Data 1). The RNA-Seq data were generated by the Genotype-Tissue Expression Consortium (GTEx v6). First, we map RNA-Seq reads to the human reference genome using a short-read aligner (performed by GTEx consortium[14]) (Fig. 1). Next, we identify reads spanning the V(D)J junction of the Ig receptors and assembled clonotypes (a group of clones with identical CDR3 amino acid sequences). We define the CDR3 as the sequence of amino acids, starting with cysteine, which is located on the left side of the junction, and ending with phenylalanine (for IGK or IGL) or tryptophan (for IGH), which is located on the right side of the junction. In this case, ImReP uses 0.02 trillion high-quality reads that had been successfully mapped to Ig genes or were unmapped reads that had failed to map to the human reference genome (Fig. 1a and Supplementary Fig. 1).

ImReP is a two-stage alignment-free approach to assembling CDR3 sequences and detecting corresponding V(D)J recombinations (Fig. 1b). In the first stage, we prepare the candidate receptor reads from mapped and unmapped RNA-Seq reads (Supplementary Fig. 1). We then merge partially mapped reads from Ig loci and unmapped reads into a set of candidate receptor reads, which serve as an input for ImReP. We scan the amino acid sequences of the read and determine the putative CDR3 as a substring of the read starting from cysteine (C) and ending with phenylalanine (F) (or tryptophan(W) for IGH). A read is separated into three parts as follows: read prefix, CDR3, and read suffix. The CDR3 sequence is a sequence starting with cysteine (C) and ending with, for IGK and IGL, phenylalanine (F), and, for IGH, tryptophan (W). Reads with putative CDR3s are further examined to assess the overlap of V and J genes. Variable Ig receptor genes are imported from IMGT version: 3.1.17. We use C from the beginning of the read and C from the V gene as an anchor to match the read prefix and the V gene. Similarly, we use F (or W) from the end of the read and F (or W) from the J gene as an anchor to match the read suffix and the J gene.

In the second stage, ImReP utilizes reads that contain a partial CDR3 sequence and overlap a single gene segment (V or J). We use an alignment-free procedure to determine the alignment between the V or J gene and the read prefix or suffix, respectively. ImReP performs matching with a suffix tree technique; matched reads with an overlap of at least 15 nucleotides are used to assemble full-length CDR3s. We further correct PCR and sequencing errors in the assembled CDR3s. ImReP clusters assemble CDR3 into a set of clusters using the CAST algorithm[10]. The clustering procedure is iteratively repeated until the average inverse edit distance (Levenshtein) inside each cluster is less than the user-defined threshold (ImReP default is .2). The consensus sequence of each cluster is reported as the correct CDR3 sequence. A detailed description of the methodology implemented with ImReP is provided in the Methods section. ImReP is freely available at https://github.com/Mangul-Lab-USC/imrep. Currently, ImReP supports human and mouse Ig receptor repertoires.

**Feasibility of using RNA-Seq to study the Ig repertoire**. To validate the feasibility of using RNA-Seq to study the Ig receptor repertoire, we simulate RNA-Seq data as a mixture of transcriptomic reads and reads derived from Ig transcripts (ratio between Ig-derived reads and transcriptomic reads was on average 1 : 3600) (Supplementary Fig. 2). Ig transcripts are simulated based on random recombination of V, D, and J gene segments (obtained from IMGT database[11]) with non-template insertion at the recombination junctions (Supplementary Fig. 3). We assess the ability of ImReP to extract CDR3-derived reads from the RNA-Seq mixture by applying ImReP to a simulated RNA-Seq mixture. While our simulation approach may not completely summarize the various nuances and eccentricities of actual immune repertoires, it allows us to assess the accuracy of our tool. ImReP is able to identify 99% of CDR3-derived reads from the RNA-Seq mixture, suggesting it is a powerful tool for profiling RNA-Seq samples of immune-related tissues.

Next, we compare ImReP with other methods designed to assemble Ig receptor repertoires. We also investigate the sequencing depth and read length required to reliably assemble Ig sequences from RNA-Seq data. Our simulations suggest that both read length and sequencing depth have a major impact on precision-recall rates of CDR3 sequence assembly. ImReP is able to maintain an 80% precision rate for the majority of simulated scenarios. Average CDR3 coverage that is higher than eight allows ImReP to archive a recall rate close to 90% for a read length above 75 bp (Fig. 2a). Increasing coverage has a positive effect on the number of assembled clonotypes achieved by ImReP.

We compare the performance of ImReP with that of MiXCR (RNA-Seq mode)[8, 13], IgBlast-based pipeline[15], and IMSEQ[13]. Except for IMSEQ, these tools were developed to assemble the hypervariable sequences from Ig receptors directly from RNA-Seq data. Another tool, iSSAKE[16], is no longer supported and was not recommended for use. Unfortunately, we obtained empty output after running V'DJer[17] and we could not solve the problem by increasing coverage in the simulated data. We exclude TRUST[9] and TraCeR[10], as those methods are solely designed for T-cell receptors. We supplied each of those tools with the original RNA-Seq reads as raw or mapped reads, depending on the software developers' recommendations. IMSEQ[13] cannot be applied directly to RNA-Seq reads because it was originally designed for targeted sequencing of Ig receptor loci. Thus, to independently assess and compare accuracy with ImReP, we only ran IMSEQ with the simulated reads derived from Ig transcripts (Supplementary Fig. 2).

ImReP consistently outperforms existing methods in both recall and precision rates. The recall is defined as $TP/(TP + FN)$. Precision is defined as $TP/(TP + FP)$. We define TP as the number of correctly assembled CDR3 sequences (based on the exact match), FN is defined as the number of true CDR3 sequences not assembled by the method and FP is defined as the number of incorrectly assembled CDR3 sequences. On average, ImReP offers three-time superior accuracy (average F-score of ImRep is .78, for other methods average F-score is < 0.2). F-score is defined as the harmonic mean of precision and recall. Notably, ImReP is the only method with acceptable performance for 50 bp read length, reconstructing with a higher precision rate significantly more CDR3 clonotypes than other methods.

To further demonstrate the feasibility of applying nonspecific RNA-Seq techniques to profile Ig receptor repertoires, we use 18 tumor biopsies sequenced by BCR-Seq and RNA-Seq. We acquired biopsies from patients with histologically confirmed Burkitt lymphoma[18]. We first mapped the reads onto the reference human genome and transcriptome, then we extracted unmapped reads, which we provided to ImReP for assembly of IGH clonotypes. Based on the recommendation of MiXCR, we provide raw paired-end reads to the tool. BCR-Seq data were generated by Adaptive Biosystems (https://www.adaptivebiotech.com/) and was analyzed by Adaptive Biosystems's Immune Analyzer package. One difficulty inherent to using BCR-Seq as a gold standard for estimating the efficiency of the RNA-Seq method is that BCR-Seq

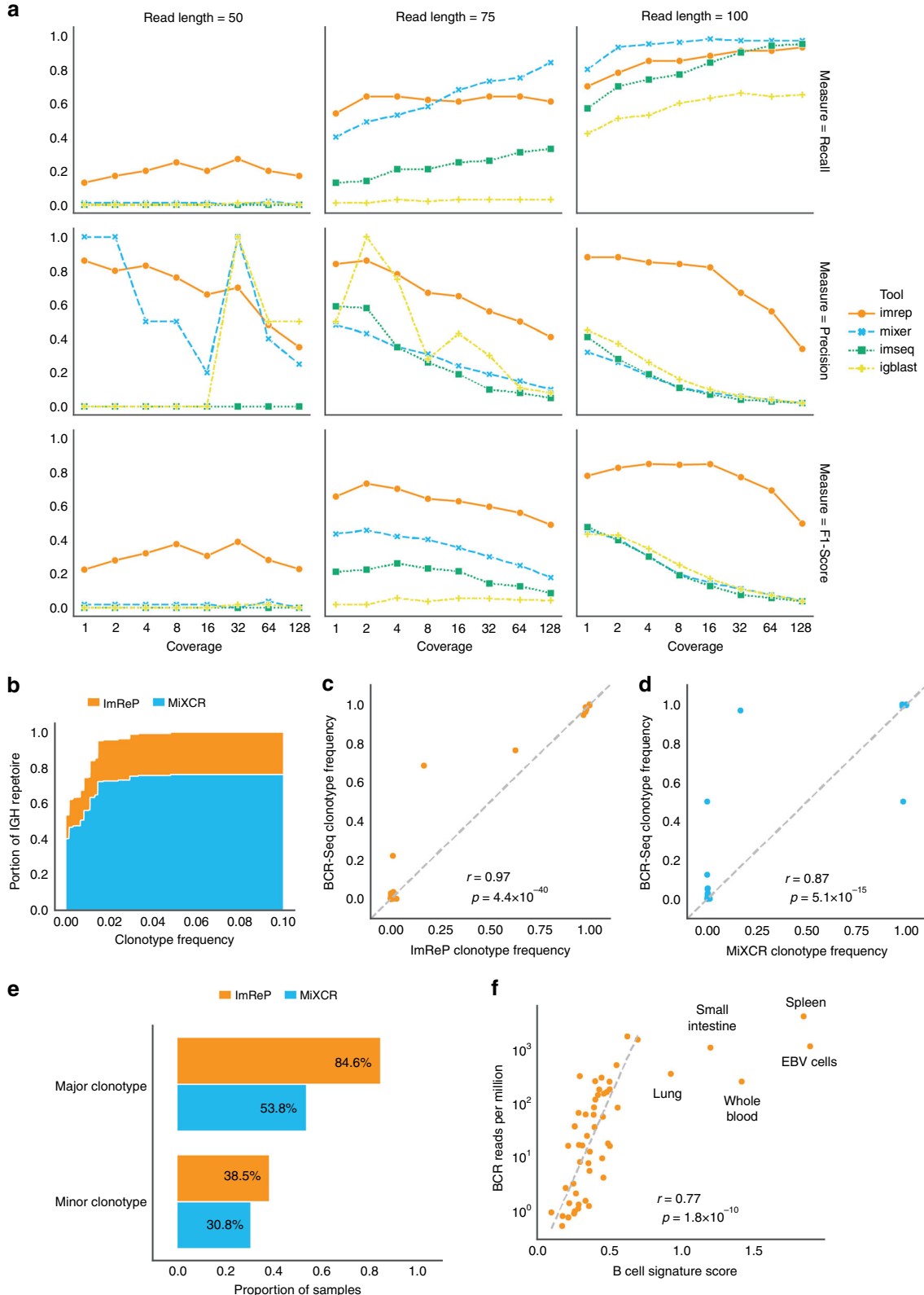

captures DNA clonotypes, whereas RNA-Seq only captures the expressed clonotypes. To account for the possible discrepancies, we first map RNA-Seq reads onto the major clonotypes with relative frequency at least 90% detected by BCR-Seq. In 5 out of 18 BCR-Seq samples, no RNA-Seq reads map to BCR-seq-confirmed major clonotypes. We exclude those samples from the

analysis. In remaining samples, we consider the set of CDR3s obtained by BCR-Seq to be the total IGH repertoire.

We investigate which portion of the total immune repertoire that RNA-Seq is capable of capturing. Using RNA-Seq, ImReP is able to capture on average 53.3% of the IGH repertoire, estimated as the sum of detected BCR-seq-confirmed clonotypes; MiXCR is

**Fig. 2 Evaluation of ImReP. a** Evaluation of ImReP based on the number of assembled CDR3 sequences and comparison with the existing methods. Precision, recall, and F-score rates for ImReP (orange), MiXCR (RNA-Seq mode) (blue), IMSEQ (green), and IgBlast (yellow) on simulated data for IGH transcripts are reported for various reads length (separate plots) and per transcript coverages (1;2;4;8;16;32;64;128) (x-axis). The recall was defined as in Eq. 3. Precision is defined as in Eq. 4. TP was defined as the number of correctly assembled CDR3 sequences (based on the exact match), FN as the number of true CDR3 sequences not assembled by the method, and FP as the number of incorrectly assembled CDR3 sequences. F-score was defined as the harmonic mean of precision and recall. **b–e** Concordance of targeted BCR-Seq and nonspecific RNA-Seq performed on 13 tumor biopsies from Burkitt lymphoma. **b** Area chart shows the proportion of the total IGH repertoire captured by ImRep (orange) and MiXCR (RNA-Seq mode) (blue), depending on the minimum BCR-seq-confirmed clonotypes frequency considered. The x-axis corresponds to BCR-seq-confirmed clonotypes frequency $Z$. The y-axis corresponds to the fraction of assembled IGH repertoire with clonotype abundances greater than $Z$. The total repertoire was defined as the sum of the BCR-seq-confirmed clonotypes abundances. **c** Pearson correlation of IGH clonotype frequencies estimated based on the BCR-Seq data (y-axis) and the RNA-seq data (x-axis) across all the samples for ImReP ($n = 63$, $r = 0.97$, p-value $= 4.4 \times 10^{-40}$). **d** Pearson correlation of IGH clonotype frequencies estimated based on the BCR-Seq data (y-axis) and the RNA-seq data (x-axis) across all the samples for MiXCR ($n = 45$, $r = 0.87$, p-value $= 5.1 \times 10^{-15}$). Only clonotypes assembled from RNA-Seq data are presented. **e** ImReP (orange) is able to detect major and minor clonotypes in a larger proportion of the samples compared with MiXCR (RNA-Seq mode) (blue). Major and minor clonotypes were defined based on BCR-Seq data as the clonotype with the largest frequency or smallest frequency, respectively. **f** Correspondence of ImReP-derived reads from Ig receptors to the relative abundance of B cells inferred across 53 GTEx tissues. Scatterplot of the number of all Ig-derived reads per 1 million RNA-Seq reads (y-axis) and B-cell signature score inferred by SaVant based on the gene expression profiles (x-axis). Pearson correlation ($n = 48$, $r = 0.77$, p-value $= 1.8 \times 10^{-10}$) is calculated excluding the five outliers labeled in the plot (Lung, Small Intestine, Whole Blood, Spleen, and EBV Cells). Source data are provided as a Source Data file.

able to capture 40.1% (Fig. 2b). In all cases, ImRep is capable of detecting BCR-seq-confirmed clonotypes with a relative frequency exceeding 90%. In comparison, MiXCR detects these clonotypes in only 83.3% of cases. When the frequency of the major clonotype drops below 10%, ImReP is able to detect the major clonotype in 60% of the cases, while MiXCR only detects a clonotype in 20% of the cases. Remarkably, both methods are able to detect major clonotypes with a frequency below 1% in one of the samples (Supplementary Data 2).

We also investigate the ability of each method to detect BCR-seq-confirmed minor clonotypes. The average frequency of the minor clonotypes across all samples is 0.37%. ImReP is able to detect a minor clonotype in 38% of the samples (Fig. 2e). Despite the ability of ImReP and MiXCR to capture the majority of BCR-seq-confirmed repertoire, both methods often miss the rare clonotypes due to the limited number of Ig-derived reads in RNA-Seq data. ImReP is able to detect 50% of all BCR-Seq-confirmed clonotypes with the relative frequency higher than 0.24%. MiXCR is able to detect 50% of all BCR-Seq-confirmed clonotypes with the relative frequency higher than 0.29% (Supplementary Fig. 4). Both methods are able to accurately estimate the relative frequencies of assembled clonotypes (ImRep: $r = 0.97$, p-value $= 4.4 \times 10^{-40}$; MiXCR $r = 0.87$, p-value $= 5.1 \times 10^{-15}$) (Fig. 2c, d). Scripts and commands utilized to process the data and run all the tools used in this study are available online at https://github.com/Mangul-Lab-USC/ImReP_publication.

We also investigate the possibility of fusing a V and J read based on the partial CDR3 overlap that was not actually derived from the same read. We obtained 3129 BCR-Seq-based IGH transcripts from a healthy, naive B-cell repertoire sequenced using error-corrected B-cell receptor (BCR) sequencing[19]. We use those transcripts as the reference to simulate reads covering the BCR-Seq-based IGH transcripts with 16× coverage. The second stage of ImReP increases sensitivity by 16% for 50 bp reads and 4% for 75 bp reads. No improvement is observed for 100 bp reads (Supplementary Table 1). The decrease of precision in case of 75 bp reads results in an overall decrease of the F-score. Based on the simulated reads, we recommend applying the second stage of ImRep for 50 bp reads (implemented as default settings in ImReP).

We further validate the ability of ImReP to accurately infer the proportion of immune cells in sampled tissue. We hypothesize that the fraction of B cells in a sample will be proportional with the fraction of receptor-derived reads in our RNA-Seq data. We use a transcriptome-based computational method, SaVant[20],

which uses cell-specific gene signatures (independent of Ig transcripts) to infer the relative abundance of B cells within each tissue sample (Supplementary Table 2). The B-cell signatures used by SaVant are derived from CD19+ cells and might not represent every B-cell subset[21]. However, CD19+ cells likely represent the largest populations of B-cell subsets and many of the CD19-negative B-cell subsets may carry a gene signature similar to the CD19 signatures. We find that B-cell signatures inferred by SaVant show a positive correlation with the size of IGH repertoire ($r = 0.77$, $P = 1.8 \times 10^{-10}$) (Fig. 2f). An exception to this correlation is found for tissues that contain the highest density of B cells: spleen, whole blood, small intestine (terminal ileum), lung, and Epstein–Barr virus (EBV)-transformed lymphocytes (LCLs).

**Characterizing the Ig repertoire across 53 GTEx tissues.** ImReP identifies over 8826 million reads overlapping 3.6 million distinct CDR3 sequences that originated from diverse human tissues. The majority of assembled CDR3 sequences were derived from IGH chain (1.7 million), 0.9 million were derived from the IGK chain, and 1.0 million were derived from the IGL chain. Ninety-eight percent of CDR3 sequences have a count of less than ten reads and the median CDR3 sequence count is 1.4. CDR3 sequences derived from IGK are the most abundant across all tissues, accounting on average for 54% of the entire B-cell population (Supplementary Fig. 5).

We compare the length and amino acid composition[22] of the assembled CDR3 sequences of Ig receptor chains (Supplementary Fig. 6). Consistent with previous studies, we observe that Ig light chains have notably shorter and less variable CDR3 lengths when compared with heavy chains[23]. The tissue type appears to have no effect on the length distribution of CDR3 sequences (Supplementary Fig. 7). In line with other studies[23, 24], both light chains exhibit a reduced amount of sequencing diversity (Supplementary Fig. 6).

We observe an average of 1331 distinct Ig clonotypes per sample. To account for various sequencing depths, we further normalized the detected number of clonotypes by the total number of RNA-Seq reads. We refer to this measure as clonotypes per one million raw RNA-Seq reads (CPM). As the number of distinct clonotypes does not increase linearly with the sequencing depth, a CPM metric should not be used in studies comparing clonotype diversity across various phenotypes. Instead, CPM is intended to be an informative measure of clonal diversity that is adjusted for sequencing depth.

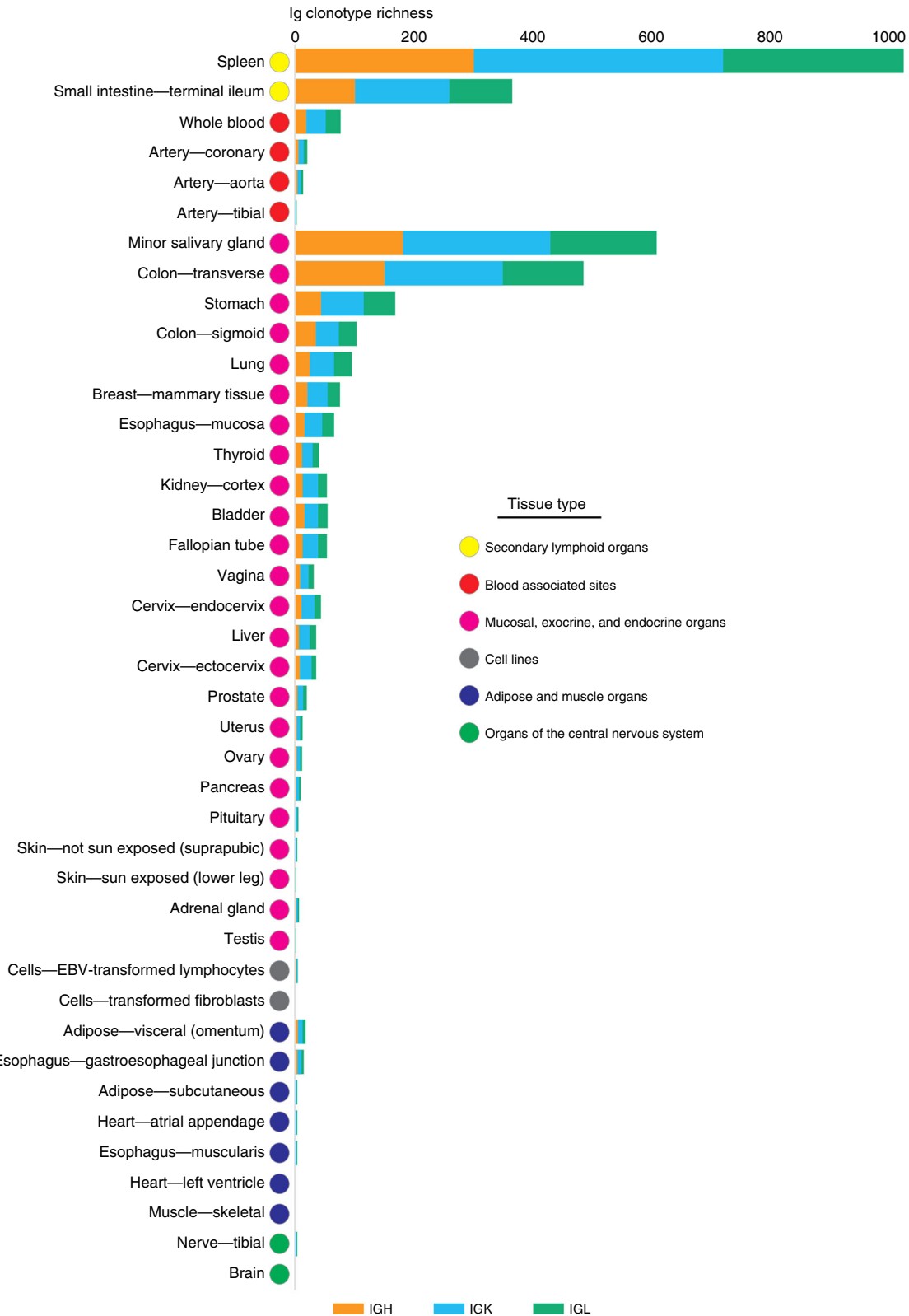

We use per sample α-diversity (Shannon entropy) to incorporate into a single diversity metric the total number of distinct clonotypes and their relative frequencies. Among all tissues, spleen has the largest B-cell population, with a median of 1301 Ig-derived reads per one million RNA-Seq reads. Spleen also has the most diverse population of B cells with median per sample α-diversity rate of 7.6, corresponding to 1025 CPM (Fig. 3 and Supplementary Data 1). Organs that possess mucosal, exocrine, and endocrine sites ($n = 24$) harbor a rich clonotype population with a median of 87 CPM per sample. Minor salivary glands have the highest Ig diversity rates in the group ($α = 7.1$) and surpass the diversity rates of the terminal Ileum containing Peyer's Patches, which are secondary lymphoid organs (Supplementary Data 1).

**Fig. 3 Adaptive immune repertoires across multiple human tissues.** Adaptive immune repertoires of 8555 samples across 544 individuals from 53 tissue types obtained from the Genotype-Tissue Expression study (GTEx v6). We grouped the tissues by their relationship to the immune system. The first group includes the lymphoid tissues ($n = 2$, yellow color). The second group includes blood associated sites including whole blood and blood vessels ($n = 4$, red color). The third group are the organs that encompass mucosal, exocrine, and endocrine sites ($n = 21$, magenta color). The fourth group are cell lines ($n = 3$, gray color). The fifth group are adipose or muscle tissues and the gastroesophageal junction ($n = 7$, blue color). The sixth group are organs from the central nervous system ($n = 14$, green color). Brain encompasses the following tissues: spinal cord (cervical c-1), amygdala, cortex, hypothalamus, caudate (basal ganglia), anterior cingulate cortex, nucleus accumbens (basal ganglia), cerebellum, frontal cortex (BA9), and cerebellar hemisphere. The histogram reports Ig clonotype richness, calculated as the number of distinct amino acid sequences of CDR3 per one million RNA-Seq reads (CPM). The median number of distinct amino acid sequences of CDR3 are presented individually for immunoglobulin heavy chain (IGH), immunoglobulin κ chain (IGK), and immunoglobulin λ chain (IGL). Source data are provided as a Source Data file.

Tissues not related to the immune system, including adipose, muscle, and organs from the central nervous system, contain a median of six CPM per sample, which are most likely due to the blood content of the tissues[25]. The highest number of distinct CDR3 sequences among non-lymphoid organs is present in the omentum, a membranous double layer of adipose tissue containing fat-associated lymphoid clusters. As expected[26], EBV-transformed lymphocytes (LCLs) harbor a large homogeneous population of Ig clonotypes (Supplementary Data 1 and Supplementary Fig. 8). The number of reported clonotypes is normalized by the proportion of B cells within each tissue sample (Supplementary Data 3). We have used SaVant to infer the relative abundance of B cells within each tissue sample based on cell-specific gene signatures (independent of Ig transcripts).

**Ig clonotypes specific to an individual or a tissue type.** Amino acid sequences of clonotypes exhibit extreme inter-individual dissimilarity, with 88% of clonotypes unique to a single individual (private) (Fig. 4a). The remaining ~400,000 clonotypes are shared by at least two individuals (public). The small fraction of B cells present in many tissues limits our ability to capture the entire Ig repertoire in those tissues and leads to mis-classification of some public clonotypes as private. The number of individuals sharing clonotypes varies across Ig chains, with Ig light chains having the highest number of public clonotypes. Twenty-five percent of all IGK clonotypes are public, and the number of individuals sharing the IGK clonotype sequences can be as high as 471 (Fig. 4b). The limited capacity of RNA-Seq to cover low-abundant clonotypes may misclassify public clonotypes as private. Consistent with previous studies[9, 27], we observe that public clonotypes are significantly shorter in length than private clonotypes (two-sided two-sample $t$-test: $p$-value $< 2 \times 10^{-16}$). For example, IGH chain public clonotypes have an average length of 13 amino acids and private clonotypes have an average length of 16 amino acids.

We also examine whether public clonotypes are more often shared across tissues than across individuals. We observe a strong correlation between the number of times receptor sequences are shared across individuals and across tissues of the same individual for both IGK ($r = 0.78$, $p$-value $< 2 \times 10^{-16}$) and IGL chains ($r = 0.77$, $p$-value $< 2 \times 10^{-16}$) (Supplementary Fig. 9). In contrast, public receptors of IGH chain are unlikely to be shared across tissues ($r = 0.15$, $p$-value $< 2 \times 10^{-16}$) (Supplementary Fig. 9). Overall, 14% of the ~240,000 clonotypes from both light and heavy chains shared across tissues are public (Fig. 4c). The full list of public clonotypes is distributed with the Atlas of Immunoglobulin Repertoires, which is publically available at https://github.com/Mangul-Lab-USC/TAIR.

**The flow of Ig clonotypes across human GTEx tissues.** A large number of samples available through this study allow us to establish a pairwise relationship between tissues and to track the flow of Ig clonotypes across various human tissues.

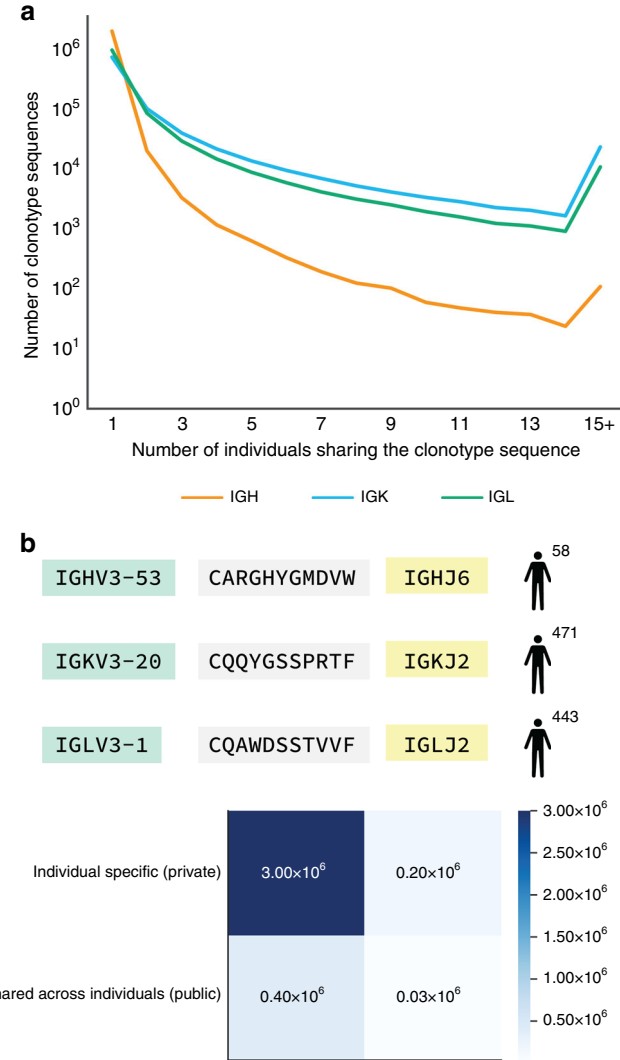

**Fig. 4 Private and public Ig clonotypes. a** Distribution of frequencies of private ($n = 1$) and public ($n > 1$) clonotypes across 544 individuals. We collect clonotypes from all tissues of the same individual and combine clonotypes into a single set corresponding to that individual. **b** The most public clonotypes (shared across the maximum number of individuals) and corresponding VJ recombination are presented for IGH, IGK, and IGL. **c** Clonotypes' sequences are classified into public (shared across individuals) clonotypes, private (individual-specific) clonotypes, tissue-specific clonotypes, and clonotypes shared across multiple tissues. The number of clonotypes falling into each pair of categories is reported. Source data for **a** and **c** are provided as a Source Data file.

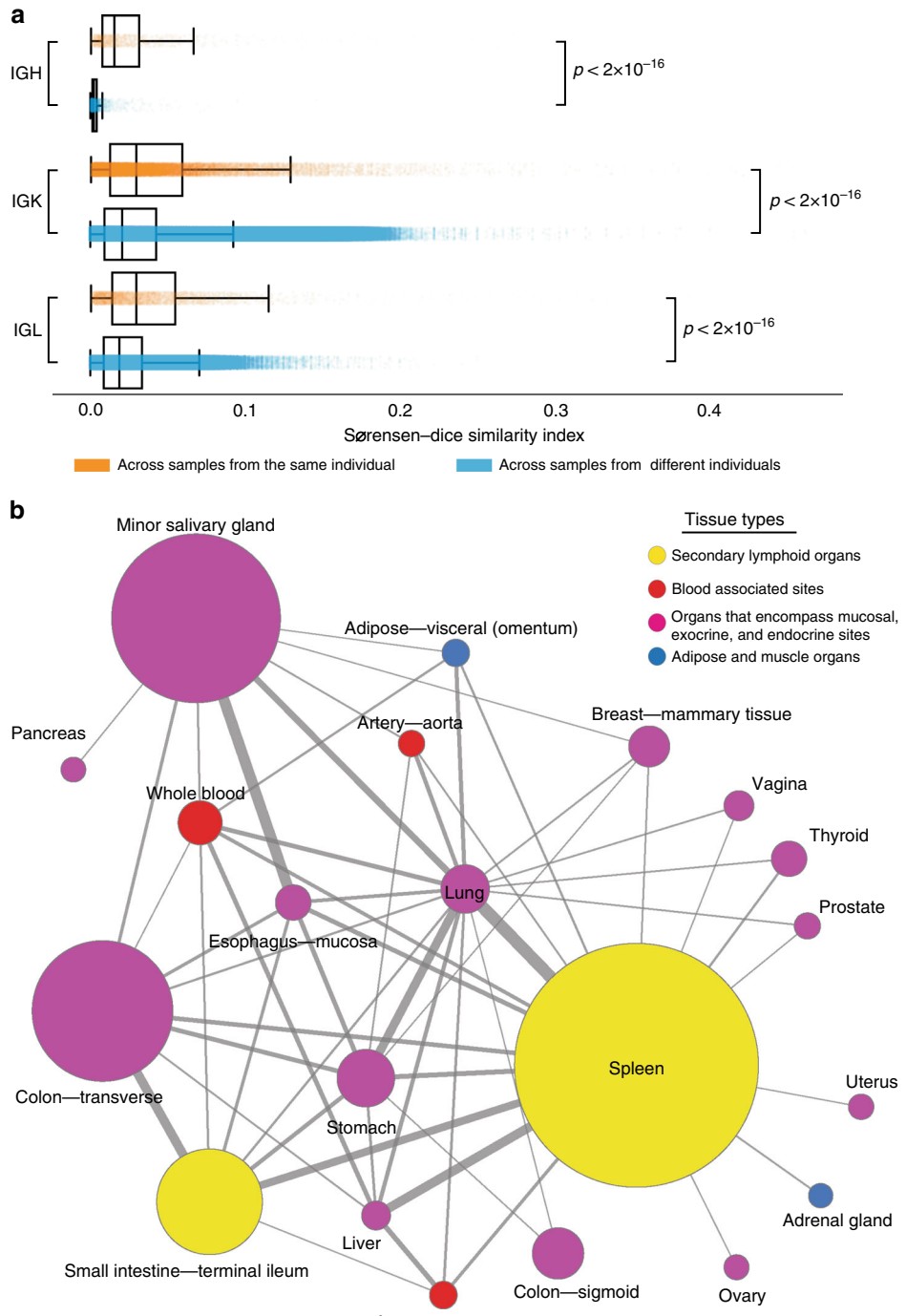

**Fig. 5 The flow of Ig clonotypes across diverse human tissues.** Results are based on pairs of tissues that are represented by at least 10 individual donors. **a** Box plots depicting the Sørensen–Dice similarity indexes for Ig clonotype sequences shared across samples from the same individual (orange color), IGH ($n = 1542$, min $= 0.0006$, $Q_1 = 0.0080$, median $= 0.0158$, $Q_3 = 0.0317$, max $= 0.2731$), IGK ($n = 30498$, min $= 0.0005$, $Q_1 = 0.0131$, median $= 0.0297$, $Q_3 = 0.0597$, max $= 0.4545$), and IGL ($n = 5124$, min $= 0.0006$, $Q_1 = 0.0142$, median $= 0.0299$, $Q_3 = 0.0548$, max $= 0.3871$); shared across samples from different individuals (blue color), IGH ($n = 2988$, min $= 0.0004$, $Q_1 = 0.0016$, median $= 0.0025$, $Q_3 = 0.0041$ max $= 0.1463$), IGK ($n = 10213560$, min $= 0.0002$, $Q_1 = 0.0091$, median $= 0.0208$, $Q_3 = 0.0424$, max $= 0.4615$), and IGL ($n = 1484735$, min $= 0.0004$, $Q_1 = 0.0090$, median $= 0.0188$, $Q_3 = 0.0336$, max $= 0.2500$). Only samples with at least ten reported clonotype sequences were used to compute Sørensen–Dice similarity indexes. Each boxplot represents the median and interquartile range, with whiskers extending to 1.5 times the interquartile range. The $p$-values were generated using a two-sided Mann–Whitney $U$-test for each chain (IGH: $U = 417655.0$, $p$-value $= 0.0$, IGK: $U = 184067549896.0$, $p$-value $= 0.0$, IGL: $U = 4864609217.7$, $p$-value $= 5.03 \times 10^{-261}$). **b** The flow of IGH clonotype across diverse human tissues is presented as a network. Each node represents a tissue; node size is proportional to the median number of clonotypes of the tissue. The color of the node corresponds to a type of the tissue: secondary lymphoid organs (yellow colors), blood associated sites (red color), and organs that encompass mucosal, exocrine, and endocrine sites (lavender color). Compositional similarities between the tissues in terms of gain or loss of CDR3 sequences are measured across valid pairs of tissues using a β-diversity (Sørensen–Dice similarity) index. Edges are weighted according to the β-diversity index. Edges with β-diversity scores > 0.001 are presented. Source data are provided as a Source Data file.

We observe a significant increase in the number of CDR3 sequences shared across pairs of tissues obtained from the same individual. Further, we consistently observe this pattern for all chains of Ig receptors (two-sided Mann–Whitney $U$-test: $p$-value $< 2 \times 10^{-16}$ for each chain) (Fig. 5a). We observe a different amount of shared CDR3 sequences across different types of Ig chains, with an increase in Ig light chains when compared to Ig heavy chains. The largest difference occurs between Ig heavy chains shared between samples taken from the same individuals (median Sørensen–Dice similarity index of 0.0158), and samples taken from different individuals (median Sørensen–Dice similarity index of 0.0025) (Fig. 5a).

To establish the flow of Ig clonotypes across various tissues, we compare clonotype populations between and within the same individuals. We limit this analysis to pairs of tissues for which we had at least 10 individuals (870 pairs of tissues out of 1378 possible pairs). We use β-diversity (Sørensen–Dice similarity index) to measure compositional similarities between the tissues in terms of gain or loss of CDR3 sequences (Fig. 5b). For the majority of the 870 available tissue pairs, we observe no commonality between IGH sequences, which corresponds to a β-diversity score of 0.0.

We examine the flow of IGH clonotypes across tissues and present it as a network (Fig. 5b). Among 870 available tissue pairs, we identify 56 tissue pairs with a β-diversity score above 0.001. The spleen has the most highly connected tissue (17 connections), followed by lung (16 connections). Clonotypes represent one connected component, meaning that every two nodes are connected either directly or via other nodes. Clonotype populations of spleen and lung are the most similar (0.02 β-diversity score). Other highly similar pairs are minor salivary gland and esophagus mucosa, as well as terminal ileum (small intestine) and transverse colon. We observe more than 200 pairs of tissues with a β-diversity score for Ig light chains above 0.001 (Supplementary Figs. 10 and 11). The most similar tissue pairs for the IGK chain are spleen and transverse colon (0.15 β-diversity score).

**ImReP identifies tissue samples with lymphocyte infiltration.** Histological images of tissue cross-sections and pathologists' notes are used to validate ImReP's ability to identify samples with a high lymphocyte content, which often correlates with a disease state. We examine the IGH clonotype populations from thyroid tissue across individuals. The median number of inferred, distinct CDR3 sequences per sample is 20, although 14.5% of the samples had more than 500 distinct CDR3 sequences. We observe the highest number of CDR3 sequences among all thyroid samples in an individual with Hashimoto's thyroiditis, an autoimmune disease characterized by lymphocyte infiltration and T-cell-mediated cytotoxicity. A text-based analysis of pathologists' notes corresponding with biological samples indicates that Hashimoto's disease was present, with varying degrees of severity, in 12.6% of examined thyroid samples. First, we use pathologists' notes to annotate samples as derived from individuals who lack a Hashimoto's disease diagnosis ($n = 180$) or were assigned a Hashimoto's disease diagnosis ($n = 26$), then we compare the adaptive repertoire diversity between the two groups. We observe a significant increase in the number of distinct IGH clonotypes in samples from individuals with Hashimoto's thyroiditis (Mann–Whitney $U$-test: $U = 83$, $p$-value $= 2.1 \times 10^{-14}$) (Supplementary Fig. 12). We also observe a significant increase in the number of distinct IGH clonotypes in positive correlation with the noted severity of Hashimoto's thyroiditis (Fig. 6a). In addition, a larger number of clonotypes in kidney samples correlates with the presence of glomerulosclerosis, and, in lung samples, a larger number of

clonotypes correlates with the presence of inflammatory diseases such as sarcoidosis and bronchopneumonia.

We observe no difference in clonal diversity in males and females across the tissue types, except in breast tissues (two-sided Mann–Whitney $U$-test: $U = 376$, $p$-value $= 4.14 \times 10^{-15}$). Higher clonotype diversity scores of breast tissue in male individuals corresponds to gynecomastia, a common disorder marked by non-cancerous enlargement of male breast tissue (Fig. 6b).

## Discussion

We develop ImReP, a computational approach capable of accurately reconstructing Ig immune repertoires using RNA-Seq data. Our initial study demonstrates the ability of ImReP to efficiently extract Ig-derived reads from RNA-Seq data and accurately assemble the corresponding hypervariable region sequences. The proposed algorithm can accurately assemble CDR3 sequences of Ig receptors, despite the presence of sequencing errors and short read length. Simulations generated using various read lengths and coverage depth show that ImReP consistently outperforms existing methods in terms of precision and recall rates.

We demonstrate the feasibility of applying RNA-Seq to study the adaptive immune repertoire. Although RNA-Seq lacks the sequencing depth of targeted sequencing (i.e., BCR-Seq), the approach compensates for these analytical restraints by examining a larger sample size. Using ImReP, we create the first systematic atlas of immune sequences for Ig receptor repertoires across diverse human tissues. This atlas provides a rich resource for comparative analysis of a range of tissue types, most of which are currently unstudied. The atlas of immune repertoires, available with the paper, is one of the largest collections of CDR3 sequences and tissue types. We anticipate that this database will enhance future studies in areas such as immunology and will contribute to the development of diagnostic tools and therapies for human disease.

Using RNA-Seq to study immune repertoires is advantageous when compared to Rep-Seq; RNA-Seq has the ability to simultaneously capture clonotype populations from all chains during a single run. RNA-Seq also allows simultaneous detection of overall transcriptional responses of the adaptive immune system, which it produces by comparing changes in the number of Ig transcripts to the much larger transcriptome. Given the increasing number of large-scale RNA-Seq datasets available, we look forward to scaling up the atlas of immune receptors in order to provide valuable insights into immune responses across various autoimmune diseases, allergies, and cancers.

## Methods

**RNA-Seq data**. We used RNA-Seq data from the Genotype-Tissue Expression study (GTEx Consortium v.6) that corresponds to 8,555 samples collected from 53 tissues that were obtained from 544 individuals. RNA-Seq data is generated using Illumina HiSeq sequencing of 75 bp paired-end reads. The data were derived from 38 solid organ tissues, 11 brain subregions, whole blood, and 3 cell lines of post-mortem donors. The samples were collected from adults matched for age across male and female individuals. Metadata for the GTEx samples used in this paper can be found in Supplementary Data 4.

**RNA-Seq data preprocessing**. We downloaded the mapped and unmapped reads in BAM format from dbGap (http://www.ncbi.nlm.nih.gov/gap). For each sample, we prepared the candidate receptor-derived reads as the input for the ImReP tool. First, we extracted reads mapped to the Ig genes. Some high-throughput aligners allow partial mapping (i.e., soft clipping), which trims one or two ends of the reads and maps the remaining read. Reads containing CDR3 sequences may be found among these reads and can be extracted using ImReP.

Second, we filter out low quality, low complexity reads and reads that match rRNA repeats. We then merged the reads mapped to the Ig loci and the prepared unmapped reads; ImReP use this data to assemble CDR3 sequences and corresponding V(D)J recombinations.

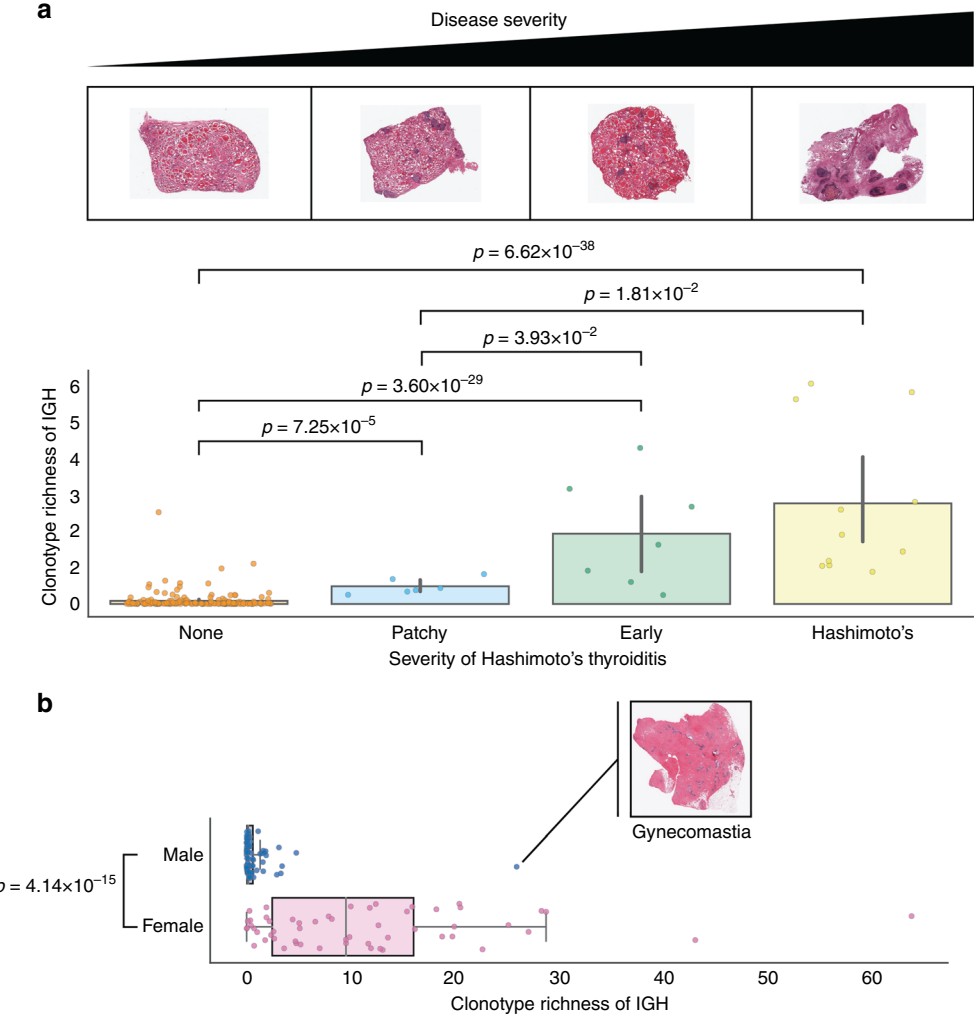

**Fig. 6 ImReP is able to identify samples with high activity of lymphocytes.** Histological images of tissue cross-sections and pathologists' notes are used to validate the ability of ImReP to detect samples with a high activity of lymphocytes. **a** Samples were ordered by the individual's severity of Hashimoto's thyroiditis, as reported by pathologists' notes: None ($n = 180$), Patchy ($n = 6$), Early ($n = 7$), and Hashimoto's ($n = 11$). Histological images are provided to illustrate each disease state. Bar plot reports clonotypic richness of IGH, calculated as the number of distinct amino acid sequences of CDR3 per one million RNA-seq reads (CPM). The $p$-values were generated using a two-sided, two-sample $t$-test for each pair of severities (None vs. Patchy: $t = -4.06$ and $p$-value $= 7.25 \times 10^{-5}$; None vs. Early: $t = -13.44$ and $p$-value $= 3.60 \times 10^{-29}$; None vs. Hashimoto's: $t = -16.31$ and $p$-value $= 6.62 \times 10^{-38}$; Patchy vs. Early: $t = -2.34$ and $p$-value $= 3.93 \times 10^{-2}$; Patchy vs. Hashimoto's: $t = -2.65$ and $p$-value $= 1.81 \times 10^{-2}$; Early vs. Hashimoto's: $t = -0.92$ and $p$-value $= 0.37$). Error bars represent the 95% confidence interval around the mean. **b** Boxplot reporting clonotypic richness of IGH, calculated as the number of distinct amino acid sequences of CDR3 per one million RNA-seq reads (CPM), in the breast tissues for males ($n = 73$, min $= 0.01$, $Q_1 = 0.09$, median $= 0.19$, $Q_3 = 0.63$, max $= 25.93$) and females ($n = 55$, min $= 0.04$, $Q_1 = 2.50$, median $= 9.56$, $Q_3 = 16.05$, max $= 63.80$). The extreme outlier among the male samples is illustrated with the histological image. Each boxplot represents the median and interquartile range, with whiskers extending to 1.5 times the interquartile range. We detect a significant difference between the clonotypic richness of IGH in the breast tissue of males and females (two-sided Mann–Whitney $U$-test: $U = 376$, $p$-value $= 4.14 \times 10^{-15}$). Source data are provided as a Source Data file.

**ImReP algorithm.** ImReP is a computational approach to assembling CDR3 sequences and detecting corresponding V(D)J recombinations from B- and T-cell receptors. ImReP consists of two stages. In the first stage, ImReP infers the CDR3 sequences from reads that simultaneously overlap V and J gene segments. We defined the CDR3 as the sequence of amino acids between the cysteine on the right of the junction and phenylalanine (for IGK or IGL) or tryptophan (for IGH) on the left of the junction. We first converted the read sequences from nucleotides to amino acids. We scanned the amino acid sequences of the read and determined the putative CDR3 as a sub-sequence of the read, starting from cysteine (C) and ending with phenylalanine (F) (and tryptophan [W] for IGH). The reads containing the described substring were considered candidate CDR3 reads. We denoted $n$ to be the length of the read. We denoted the coordinates of the putative CDR3 string to be $x$ and $y$, corresponding with the start and the end of the CDR3 sequence, respectively, in the read coordinates. This way each candidate CDR3 read is composed of three parts. The first part of the CDR3 read contains a prefix of the read, which potentially overlaps with the suffix of V gene. The prefix

contains the amino acids from the read, from position 0 to $x - 1$. The second part of the CDR3 read is a substring of the read containing the putative CDR3 sequence. It contains the amino acids from the read, from position $x$ to $y$. The third part of the CDR3 is a suffix of the read that potentially overlaps with the prefix of J gene. The suffix contains the amino acids of the read, from position $y + 1$ to $n$.

The amino acid sequences of the V and J genes of BCRs were imported from IMGT information system (http://www.imgt.org/vquest/refseqh.html#V-D-J-C-sets). For each V gene, we identified the last conserved cysteine (C) and recorded the position $p_C$ of the read. For each J gene, we identified the first conserved phenylalanine (for IGK or IGL) or tryptophan (for IGH) and recorded its position $p_F$. (We identified the position of phenylalanine or tryptophan in the J gene sequence. Such positions are referred to as $p_F$.) For each V gene, we extracted two substrings:

$$V_x = V[0, p_C - 1] \text{ and } V_y = V[p_C + 1, n_V] \quad (1)$$

For each J gene, we recorded two substrings:

$$J_x = J[0, p_F - 1] \text{ and } J_y = J[p_F + 1, n_J] \quad (2)$$

Here, $n_V$ and $n_J$ are the lengths of V and J genes, respectively. Given a set of candidate CDR3 reads, we attempted to find the corresponding V and J genes. We matched a substring of the read $r[0, x - 1]$ with the corresponding suffix of $V_x$ for V genes. We also matched the read $r[y + 1, n]$ with the corresponding prefix of $J_x$ for J genes. We considered a read to match the V gene if the length of $r[0, x - 1]$ is greater than four and the edit distance between $r[0, x - 1]$ and $V_x$ is <2. We considered a read to match the J gene if the length of $r[y + 1, n]$ is >4, and the edit distance between $r[y + 1, n]$ and $J_x$ is <2. In cases where a read overlaps equally (in terms of edit distance) among multiple V genes and J genes, all matching V genes are reported.

In the second stage, ImReP utilizes the reads overlapping only with the V or J gene. Such reads contain a partial CDR3 sequence. ImReP builds a suffix tree $S$ on the reads overlapping any of the V genes. Then, for each read $j$ overlapping a J gene a V-gene overlapping read, $v$ from $S$ is determined (in cases where any exists). Reads $v$ and $j$ are concatenated (based on the overlap) and the CDR3 region is extracted.

Further, ImReP uses a CAST clustering technique to correctly assemble CDR3s for PCR and sequencing errors. The output of the algorithm is the set of CDR3 partitions, and each of the partitions corresponds to a clonotype. Specifically, ImReP builds a complete graph $G = (V, E, w)$, where the set of vertices $V$ is represented by the set of assembled CDR3 sequences. The weight of the edge is determined by the inverse of the edit distance, computed between the two CDR3 sequences $x$ and $y$. The CAST algorithm is executed with the following procedure. A new partition $P$ is initialized with the max-degree node. Then, the set of "close" vertices is iteratively added to the partition, and the set of "distant" vertices are removed from the partition. A vertex $v$ is deemed to be "close" ("distant"), if the average distance from $v$ to the vertices from $P$ is greater (smaller) than a user-defined threshold. The procedure is repeated until either the set of "close" or the set of "distant" vertices is empty. In such a way, the partition $P$ is based on a max-degree node and extended with the "close" vertices. Vertices belonging to $P$ are then removed from the graph $G$ and the clustering procedure is repeated until all of the vertices are assigned to a partition. Let $\{v_1, v_2, \ldots, v_i, \ldots, v_n\}$ be a partition output by the CAST algorithm. Each $v_i$ has an associated weight equal to the count of CDR3's $v_i$, which was assembled during the first two stages of ImReP. We computed the weighted consensus sequence of $P$ and output the sequence as a final clonotype. Finally, we mapped D genes (for IGH) onto assembled CDR3 sequences and infer corresponding V(D)J recombination. Starting with release v0.8, ImReP reports the out-of-frame CDR3 sequences.

**Validation based on simulated RNA-Seq data**. We performed in-silico simulations to investigate the feasibility of using RNA-Seq to study the adaptive immune repertoire. We first checked the ability of ImReP to extract the receptor-derived reads from raw RNA-Seq reads. First, we simulated the Ig transcripts, which are composed of recombined VDJ segments containing non-template insertion at the V(D)J junction (Supplementary Fig. 2). We used the IMGT database (http://www.imgt.org/vquest/refseqh.html) of V and J gene segments. We randomly selected V, D, and J segments, and we inserted a sequence of random nucleotides between V and D, and between D and J. The length of the inserted sequence was sampled from the Gaussian-like distribution with a mean value of 15. We also excluded the simulated transcripts that contain random insertions leading to out-of-frame proteins. We used LymAnalizer (version 1.2.2) (https://sourceforge.net/projects/lymanalyzer/) to validate CDR3 sequences of the transcript. We used SimNGS (version 1.6) (https://www.ebi.ac.uk/goldman-srv/simNGS/) to simulate paired-end reads, referred as receptor-derived reads, from Ig transcripts. Next, we simulated 50 million transcriptomic reads from a human transcriptome reference (GRCh37). We mixed receptor-derived reads with transcriptomic reads into an RNA-Seq mixture (Supplementary Fig. 3). We then applied ImReP to a simulated RNA-Seq mixture in order to check the ability of ImReP to extract CDR3-derived reads from the RNA-Seq mixture.

Next, we studied the effects of the coverage and read length on the ability to reconstruct CDR3 sequences. In total, we simulated 1,000 Ig transcripts. We simulated paired-end reads of various read lengths ($l = 50, 75, 100$). We have also simulated different numbers of reads that correspond to different coverage rates of Ig transcripts ($c = 1, 2, 4, 8, 16, 32, 64, 128$). We used the power law distribution to assign frequencies to simulated Ig transcripts[28]. The CDR3 amino acid sequences assembled by ImReP were compared to simulated transcripts in order to evaluate the recall and precision for various read lengths and coverage rates.

We define recall and precision in the following way:

$$\text{Recall} = TP/(TP + FN) \quad (3)$$

$$\text{Precision} = TP/(TP + FP) \quad (4)$$

Where TP is the number of correctly assembled CDR3 sequence features (i.e., an exact match to the simulated CDR3), FN is the number of simulated CDR3 sequence features not assembled by the method, and FP is the number of incorrectly assembled CDR3 sequences. Scripts that simulate the reads and Ig

transcripts are available online at: https://github.com/Mangul-Lab-USC/ImReP_publication.

**Validation based on BCR-Seq-based IGH transcripts**. Additionally, we used Ig sequences assembled from targeted BCR-Seq data, which was derived from the IGH locus spanning the region between the FR1 to the IGHJ gene. Error-corrected BCR sequencing was used to generate sequences from peripheral blood mononuclear cells that were sampled from a healthy individual using[19]. Targeted BCR amplification was performed using a two-step RT-PCR protocol with multiplex IGHV gene primers and a barcoded IGHJ primer. Amplicons were sequenced using MiSeq 300 bp paired-end libraries and following Illumina protocols. Raw sequence reads are available under the EGAN00001419382 accession number in the European Genome-Phenome Archive. BCR-Seq was approved by the Wellcome Sanger Institute review boards and ethics committees (07/MRE05/44). As the Ig sequences constitute human data, they are stored under managed data access according to the Wellcome Trust data release policy. Access to these samples must be requested from the Data Access Committee (DAC), whose contact details can be found on the EGA study page. Data hosted on DAC have an accession number; more information can be obtained by sending an email to datasharing@sanger.ac.uk. The requester will be required to sign a data access agreement, which is in place to protect the identity of the sample donor via a managed access system.

We ran the online version of IgBLAST using the default alignment options (https://www.ncbi.nlm.nih.gov/igblast/igblast.cgi) to extract the CDR3 sequence from each Ig transcript. The resulting 3129 sequences fall into 435 distinct CDR3 sequences. Next, we used simLibrary (version 1.3) and simNGS (version 1.6) software packages to simulate three Illumina single-end read datasets at 16x coverage with read lengths of 50 bp, 75 bp, and 100 bp. The commands for this process used can be found on Github: https://github.com/Mangul-Lab-USC/ImReP_publication.

On each of the datasets, we ran ImReP with the default options and the "–noOverlap" option.

**Determine parameters for clustering using the CAST algorithm**. To cluster the assembled clones with a good balance between sensitivity and precision, we determined the threshold for inverse edit distance used by CAST. The inverse edit distance is iteratively used by CAST inside each cluster until the average inverse edit distance (Levenshtein) inside each cluster is less than the user-defined threshold. In general, the decrease of sensitivity shows that many true positive clones are collapsed by CAST. he increase in precision shows the advantage of CAST application, where false positive clones are collapsed. The CAST's threshold for edit distance was set to maximize F-score across different coverages and read lengths. Using the simulated data, we set up the defaults of CAST's threshold for an edit distance to 0.25 (Supplementary Fig. 13). We simulated paired-end reads of length $2 \times 75$ bp, covering the Ig transcript with an average coverage rate of 8×. After the CAST threshold was determined, we applied ImRep with and without CAST on simulated datasets with various read lengths and coverages (Supplementary Fig. 14). Across the vast majority of read lengths and coverages, CAST provides an improved clonotype reconstruction accuracy (measured by F-score). Lower edit distance threshold (<0.1) results in drop of sensitivity, suggesting that closely related but bona fide distinct mutants are collapsed by the algorithm. The default edit distance threshold chosen by ImRep balances both precision and sensitivity; the default setting avoids collapsing closely related-yet-bona-fide distinct clonotypes.

**Comparison with other methods**. We used simulated and real datasets to compare ImReP to existing methods. We note that IMSEQ cannot be applied to RNA-Seq reads, because it was originally designed for BCR-Seq. Scripts and commands utilized to repertoire assembly tools are available online at: https://github.com/Mangul-Lab-USC/ImReP_publication.

**Cell-type composition**. B-cell signature values per sample were derived using SaVant[20]. Cell-specific signature genes are first defined from a set of cells/tissues obtained from the Human Body Atlas[29] by using the proportional median values. We calculate these values by dividing the intensity of a probe in a particular cell type by its median value across all cells/tissues. The top 25 genes with the highest proportional median value for CD19+ B cells were defined as the specific signature for that cell type (Supplementary Table 2). All Ig genes were removed from the signature. The signature score is then generated from the average of the log2-transformed values of the signature genes within each sample.

**Definition of clonotype**. Clonotypes are defined as clones with identical CDR3 amino acid sequences.

**Histological images and pathologist notes**. We used histological images and pathologists' notes (available from the GTEx portal, http://www.gtexportal.org/home/histologyPage#data) to validate the adaptive immune profile of the samples. Although samples were derived from primary tissues, they often have a mixed cell type composition. For example, samples from stomach tissues have various

proportions of lymphocytes as, according to pathologists' notes, they were derived from mucosal or muscularis areas of the tissue. GTEx samples with inflammation and/or subject to various diseases are separately investigated. Pathologists' notes report the percentage of mucosa, and the disease or inflammation status, of the biopsied tissue.

**Statistics and reproducibility**. All statistics have been derived where appropriate sample size dictated that a statistical test could be performed.

**Data representation**. We used WebLogo3 (http://weblogo.threeplusone.com/manual.html) to visualize the amino acid composition of assembled CDR3 sequences, and Gephi (https://gephi.org/users/) to visualize the flow of clonotypes across diverse human tissues.

**Reporting summary**. Further information on research design is available in the Nature Research Reporting Summary linked to this article.

## Data availability
All RNA-Seq data discussed in this paper is available as part of the Genotype-Tissue Expression (GTEx) Project under the phs000424.v8.p2 accession number in the database of Genotypes and Phenotypes (dbGaP). The targeted BCR-Seq assemblies used for validation of the ImReP method are available at the adaptive biosystems webpage (https://clients.adaptivebiotech.com/pub/lombardo-2017-bloodadvances). RNA-Seq samples used for validation of ImReP are available under the SRP099346 accession number in the Sequence Read Archive. Raw sequence reads of targeted BCR-Seq data are available under the EGAN00001419382 accession number in the European Genome-Phenome Archive. All data required to produce the figures and analysis performed in this paper are freely available at https://github.com/Mangul-Lab-USC/ImReP_publication and are available in the Source Data zip file, including the data used to produce Figs. 2a–f, 3, 4a, c, 5a, b, and 6a, b, and Supplementary Figs. 4, 5, 6d, 7, 8, 9a–c, 10–12, 13a–c, and 14a–c.

## Code availability
ImReP is freely available at https://github.com/Mangul-Lab-USC/imrep. ImReP is distributed under the terms of the General Public License version 3.0 (GPLv3). All code required to produce the figures and analysis performed in this paper are freely available at https://github.com/Mangul-Lab-USC/ImReP_publication. Source data are provided with this paper.

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

## Acknowledgements
We thank Dr. Lana Martin for the helpful discussions and comments on the manuscript, and for assistance with producing the figures.

## Author contributions
S.M. designed the study. I.M. developed the methods and the simulated datasets used in this study. B.S., D.Y., J.R., J.R.R., H.T.Y., V.P., and W.V.D.W. performed analysis for this paper. B.S., D.Y., H.T.Y., I.M., J.R., V.P., and W.V.D.W. generated figures. A.Z., D.M., E.E., I.M., K.M.A., M.R., N.S., N.Z., R.S., S.M., S.S., and V.P. wrote the paper. A.Z., D.M., E.E., K.M.A., M.R., N.S., N.Z., R.S., S.M., S.S., and V.P. reviewed and edited the manuscript.

## Competing interests
The authors declare no competing interests.
