## [Peer Review File · Nature Communications]

Editorial Note: This manuscript has been previously reviewed at another journal that is not operating a transparent peer review scheme. This document only contains reviewer comments and rebuttal letters for versions considered at Nature Communications .

Reviewers' comments:

Reviewer #1 (Remarks to the Author):

"Profiling immunoglobulin repertoires across multiple human tissues by RNA sequencing" represents a substantial revision of a previous submission by these authors. A new validation set is included (on Burkitts lymphoma) and the previous data on TCR repertoires that had raised multiple concerns has been removed entirely. As a result, the manuscript is significantly improved, but a few specific concerns remain.

1) In my original critique, I expressed concern about the possibility of the overlap between two N-terminal and C-terminal (or 5' and 3') segments of the CDR3 representing a "false positive"—that is that ImREP would fuse a V and J read based on the partial CDR3 overlap that were not actually derived from the same read. This should be a reportable number from their simulated data—can they tell us what it is?

2) Relatedly, the authors perform an error correction step (which is particularly difficult for BCRs due to somatic hypermutation) which is based on clustering using the CAST algorithm, repeated Levenshtein distances are below a threshold—how was this threshold determined? How do the authors avoid clustering closely related but bona fide distinct mutants? Similar to comment 1, was any benchmarking done on different types of BCR data to determine how reasonable this was?

3) The validation set is on Burkitts lymphoma (a B cell tumor) samples—are the CDR3s from the tumor cells themselves? This might be clarified in the text

4) In figure 2e, what data are the non-tumor tissue coming from? This isn't still the Burkitts samples is it? What is this data source?

5) The authors make the claim that public responses (somewhat loosely defined as being in more than one person)—were not more likely to be shared across tissues. Can the authors put stats on this? If they used a more rigorous definition of public, does that conclusion change? I suspect—particularly for the light chains—that the most public receptors are also more likely to be found across tissues...maybe a correlation analysis would be helpful here.

6) In Figure 6a, the clonotype numbers are described as "Averages" but no std deviation or errors are provided and no stats are listed. It's also inappropriate to plot this as a continuous line. How many individuals are in each group? Are they statistically different from one another?

7) There are several grammatical issues throughout along with other typos—a professional editor should be consulted. Some quick examples—In the second to last sentence of the abstract, "recourse" should be "resource". The sentence on page 27 starting "Clonotypes represents..." is grammatically incorrect (I think...should be "Clonotype represents"? but I also don't know what the sentence means.

8) Minor point: At the top of page 5, the authors state "existing methods that are capable of assembling Ig repertoire sequences produce results with low accuracy"....they should specify "from bulk RNA-Seq data, relative to ImREP".

9) Minor point: the authors have called their (potentially useful) atlas the "AIR"—one of the major groups establishing standards for reporting repertoire data is already called "AIRR" and has published several reports using that title—there's a possibility for some confusion here.

Response to Reviewers' Comments

Reviewer 1 Remarks to the Author:

“Profiling immunoglobulin repertoires across multiple human tissues by RNA sequencing” represents a substantial revision of a previous submission by these authors. A new validation set is included (on Burkitts lymphoma) and the previous data on TCR repertoires that had raised multiple concerns has been removed entirely. As a result, the manuscript is significantly improved, but a few specific concerns remain.

Reviewer 1 Specific Comment #1:

In my original critique, I expressed concern about the possibility of the overlap between two N-terminal and C-terminal (or 5' and 3') segments of the CDR3 representing a “false positive”—that is that ImREP would fuse a V and J read based on the partial CDR3 overlap that were not actually derived from the same read. This should be a reportable number from their simulated data—can they tell us what it is?

Author Response:

In response to the reviewer’s query, we performed additional experiments to determine the number of false positives that result from merging overlapping reads across different read lengths by the second stage of ImRep . We added to the manuscript the following text reporting on our results:

“We have also investigated the possibility of fusing a V and J read based on the partial CDR3 overlap that was not actually derived from the same read. We have obtained 3129 BCR-Seq-based IGH transcripts from a healthy naive B cell repertoire sequenced using error-corrected BCR sequencing (Petrova et al. 2018). We used those transcripts as the reference to simulate reads covering the BCR-Seq-based IGH transcripts with 16x coverage. The second stage of ImReP increases sensitivity by 16% for 50bp reads, and 4% for 75bp reads. No improvement was observed for 100bp reads (Table S7). The decrease of precision in case of 75bp reads resulted in an overall decrease of F-score. Based on the simulated reads, we have recommended applying the second stage of ImRep for 50bp reads (implemented as default settings in ImReP). “

We have added to the supplementary materials for the manuscript the following table reporting on our results:

	Read length	Sensitivity	PPV	F-Score	TP	FP	TN
Overlap	50	21.0%	84.0%	33.6%	92	17	343
	75	25.0%	90.0%	39.1%	108	12	327
	100	26.0%	88.0%	40.1%	112	15	323
No	50	5.0%	80.0%	9.4%	20	5	415

overlap	75	21.0%	92.0%	34.2%	92	8	343
	100	26.0%	92.0%	40.5%	112	10	323

Table S7. The effect of the overlap algorithm (2nd stage of ImRep) on the accuracy of assembled Ig clonotypes. IGH transcripts were obtained by targeted BCR-Seq (see **Section of the Supplementary Materials** ‘Validation based on BCR-Seq-based IGH transcripts’). Reads were generated from BCR-Seq-based transcripts using simNGS (<https://www.ebi.ac.uk/goldman-srv/simNGS/>).

Reviewer 1 Specific Comment #2:

Relatedly, the authors perform an error correction step (which is particularly difficult for BCRs due to somatic hypermutation) which is based on clustering using the CAST algorithm, repeated Levenshtein distances are below a threshold—how was this threshold determined? How do the authors avoid clustering closely related but bona fide distinct mutants? Similar to comment 1, was any benchmarking done on different types of BCR data to determine how reasonable this was?

Author Response:

We thank the reviewer for this important point. We have performed additional experiments to identify any potential effects of the default threshold for clustering using the CAST algorithm (0.2) on both precision and sensitivity. Lower edit distance threshold (<.1) results in drop of sensitivity, suggesting that closely related but bona fide distinct mutants are collapsed by the algorithm. The defaults edit distance threshold chosen in ImRep balance both precision and sensitivity and avoids collapsing closely related but bona fide distinct clonotypes. We have added to the Supplementary Method Section of the manuscript a new section reporting our results:

“Determine parameters for clustering using the CAST algorithm.

We have first determined the threshold for inverse edit distance used by CAST to cluster the assembled clones with a good balance between the sensitivity and precision. The inverse edit distance is iteratively used by CAST inside each cluster until the average inverse edit distance (Levenshtein) inside each cluster is less than the user-defined threshold. In general, the decrease of sensitivity shows that many true positive clones are collapsed by CAST, the increase in precision shows the advantage of CAST application, where false positive clones are collapsed. The CAST’s threshold for edit distance was set to maximize F-score across different coverages and read lengths. Using the simulated data from “Validation based on simulated RNA-Seq data” Section, we have set up the defaults CAST’s threshold for edit distance to 0.25 (**Figure S14**). We simulated paired-end reads of length 2x75bp, covering the Ig transcript with average coverage 8x. After the CAST threshold was determined we have applied ImRep with and without CAST on simulated datasets with various read lengths and coverages (**Figure**

S15). Across the vast majority of read lengths and coverages scenarios CAST provides an improved clonotype reconstruction accuracy (measured by F-score).”

In addition, the following figures have been added to the manuscript:

(a)

(b)

(c)

Figure S14. The effect of the edit distance threshold used by CAST algorithm on the precision (a), sensitivity (b), and F-score (c). Ig transcripts were simulated based on the random recombination of V and J gene segments (IMGT database) with non-template insertion at the recombination junction (see **Supplementary Materials** ‘Validation based on simulated RNA-Seq data’ Section). We simulated paired-end reads of length 2x75bp, covering the Ig transcript with an average coverage rate of 8x.

(a)

(b)

(c)

Figure S15. The effect of the CAST clustering algorithm on the accuracy of assembled Ig clonotypes. Ig transcripts were simulated based on the random recombination of V and J gene segments (IMGT database) with non-template insertion at the recombination junction (see **Supplementary Materials** ‘Validation based on simulated RNA-Seq data’ Section). F-score rates for ImReP with CAST (blue) and ImReP without CAST (orange) on simulated data for immunoglobulin heavy (IGH) transcripts are reported for various reads lengths—50bp **(a)**, 75bp **(b)**, and 100bp **(c)**—and per transcript coverage rates—1,2,4,8,16,32,64,128.

Reviewer 1 Specific Comment #3:

The validation set is on Burkitts lymphoma (a B cell tumor) samples—are the CDR3s from the tumor cells themselves? This might be clarified in the text

Author Response:

All the samples used in this study were obtained from a cohort of patients diagnosed with Burkitts lymphoma. We modified text in the paper to clarify that the CDR3s were directly obtained from the tumor cells via biopsy:

“To further demonstrate the feasibility of applying non-specific RNA Sequencing to profile *Ig* receptor repertoires, we have used 18 tumor biopsies sequenced by BCR-Seq and RNA-Seq. Biopsies were acquired from the patients with histologically confirmed Burkitt lymphoma.”

Reviewer 1 Specific Comment #4:

In figure 2e, what data are the non-tumor tissue coming from? This isn't still the Burkitts samples is it? What is this data source?

Author Response:

We thank the reviewer for pointing out this confusion aspect of our manuscript. The results in Figure 2e were obtained based on the GTEx data. We modified the following text in the caption of Figure 2 for clarity:

“Correspondence of ImReP-derived reads from *Ig* receptors to the relative abundance of B cells inferred across 53 GTEx tissues. Scatterplot of the number of all *Ig*-derived reads per 1 million RNA-Seq reads (y-axis) and B-cell signature score inferred by SaVant based on the gene expression profiles (x-axis).”

Reviewer 1 Specific Comment #5:

The authors make the claim that public responses (somewhat loosely defined as being in more than one person)—were not more likely to be shared across tissues. Can the authors put stats on this? If they used a more rigorous definition of public, does that conclusion change? I suspect—particularly for the light chains—that the most public receptors are also more likely to be found across tissues...maybe a correlation analysis would be helpful here.

Author Response:

We thank the reviewer for this important comment. We have performed correlation analysis and indeed for both IGL and IGK chains the most public receptors are also more likely to be found across tissues. We added to the manuscript the following text:

“We also examined whether the public clonotypes were more often shared across tissues within an individual. Public receptors of light chains were more likely to be

shared across tissues. We observed a strong correlation between the number of times receptor sequences were shared across individuals and across tissues of the same individual for both IGK ($r = 0.78$, $p\text{-value} < 2 \times 10^{-16}$) and IGL chains ($r = 0.77$, $p\text{-value} < 2 \times 10^{-16}$) (**Figure S9**). In contrast public receptors of IGH chain were unlikely to be shared across tissues ($r = 0.15$, $p\text{-value} < 2 \times 10^{-16}$) (**Figure S10**). Overall 14% of the ~240,000 clonotypes from both light and heavy chains shared across tissues were public (**Figure 4c**).”

Figure S4 and S5 were added to the manuscript’s Supplemental Materials:

(a)

(b)

Figure S9. Scatter plot depicting the number of times receptor sequences were shared across tissues (x-axis) and across individuals (y-axis). **(a)** IGK **(b)** IGL

Figure S10. Scatter plot depicting the number of times receptor sequences of IGH chain were shared across tissues (x-axis) and across individuals (y-axis).

Reviewer 1 Specific Comment #6:

In Figure 6a, the clonotype numbers are described as “Averages” but no std deviation or errors are provided and no stats are listed. It’s also inappropriate to plot this as a continuous line. How many individuals are in each group? Are they statistically different from one another?

Author Response:

We thank the reviewer for this important comment. We have verified the number of individuals per disease group. Group with ‘late stage hashimotos’ contained only one sample and therefore was merged with ‘hashimotos’ group. Group ‘focal hashimotos’ contained two samples was also excluded from the disease group analysis. Updated figure 6a as a barplot (levels of significance between the disease group are flagged with an asterisk) is provided below:

Figure 6. ImReP is able to identify samples with high activity of lymphocytes. Histological images of tissue cross-sections and pathologists' notes have been used to validate the ability of ImReP to detect the samples with a high activity of lymphocytes. **(a)** Samples were ordered by the individual's severity of Hashimoto's thyroiditis, as reported by pathologists' notes. Histological images are provided to illustrate the disease state. Bar plot depicting the number of Ig clonotypes per sample across disease group. **(b)** Boxplot reporting number of clonotypes in the breast tissues for males and females. The outlier among the male samples is illustrated with the histological image.

We have also compared the difference in number of clonotypes between the disease groups. We added the following text to the '**ImReP identifies tissue samples with lymphocyte infiltration**' section of the paper (modified text is presented in red):

"We observed a significant increase in the number of distinct IGH clonotypes in samples from individuals with Hashimoto's thyroiditis (p-value= 1.5×10^{-5}) (**Figure S13**). **A significant increase in the number of distinct IGH clonotypes was also observed as the severity of Hashimoto's thyroiditis was increasing (Figure 6a).**"

Reviewer 1 Specific Comment #7:

There are several grammatical issues throughout along with other typos—a professional editor should be consulted. Some quick examples—In the second to last sentence of the abstract, "recourse" should be "resource". The sentence on page 27 starting "Clonotypes represents..." is grammatically incorrect (I think...should be "Clonotype represents"? but I also don't know what the sentence means.

Author Response:

We acknowledge that our past manuscript would have benefited from a professional proofreader. As was suggested, we worked with a professional editor on the revised manuscript in order to correct grammatical errors and improve clarity and flow.

Reviewer 1 Specific Comment #7:

Minor point: At the top of page 5, the authors state "existing methods that are capable of assembling Ig repertoire sequences produce results with low accuracy"....they should specify "from bulk RNA-Seq data, relative to ImREP".

Author Response:

Thank you; we modified the text to clarify the specific context in which ImReP is most accurate: "Existing methods that are capable of assembling Ig repertoires from bulk RNA-Seq data produce low-accuracy results (f-score<.2) relative to ImReP (**Figure 2a**)."

Reviewer 1 Specific Comment #7:

Minor point: the authors have called their (potentially useful) atlas the “AIR”—one of the major groups establishing standards for reporting repertoire data is already called “AIRR” and has published several reports using that title—there’s a possibility for some confusion here.

Author Response:

We thank the reviewer for pointing out the existing use of this acronym. To avoid confusion with AIR, we changed the name of our atlas to “TAIR.”

REVIEWERS' COMMENTS:

Reviewer #1 (Remarks to the Author):

Most of my concerns have been well-addressed, however in Figure 6a, rather than box plots, individuals should be plotted as points (which can then include box and whisker plots). The number of individuals in each category is not provided in the figure or the legend, which is the minimal standard for NPG journals as I understand it. This is marked as "confirmed" when it is not the case: Individual data points are shown when possible, and always for $n \geq 10$

REVIEWERS' COMMENTS:

Reviewer #1 (Remarks to the Author):

Most of my concerns have been well-addressed, however in Figure 6a, rather than box plots, individuals should be plotted as points (which can then include box and whisker plots). The number of individuals in each category is not provided in the figure or the legend, which is the minimal standard for NPG journals as I understand it. This is marked as "confirmed" when it is not the case:

Individual data points are shown when possible, and always for $n \geq 10$

Authors' Response: We have changed Figure 6a to better represent the data, and updated the figure legend. In addition to changes requested by the Editor, we corrected grammatical errors and edited sentence and paragraph structure for improved clarity and flow.